# Screening for Post-Stroke Cognitive Impairment in Sub-Saharan Africa: A Good IDEA?

**DOI:** 10.3390/brainsci15060543

**Published:** 2025-05-22

**Authors:** Fode Abass Cissé, Yannick Fogoum Fogang, Male Dore, Gilles Naeije

**Affiliations:** 1Neurology Department, Ignace Deen Hospital, 9^e^ Boulevard, Kaloum, Conakry BP 5676, Guinea; abass3002@yahoo.fr (F.A.C.); doreramatoulaye@gmail.com (M.D.); 2Neurology Department, Bafoussam Regional Hospital, Marché B, 237 Route de Bamenda, Bafoussam BP 0000, Cameroon; yanfogang@yahoo.fr; 3Stroke Unit, HUB-Hôpital Erasme, Route de Lennik 808, 1070 Bruxelles, Belgium

**Keywords:** stroke, post-stroke cognitive impairment, sub-Saharan Africa, IDEA, epidemiology

## Abstract

Background: Post-stroke cognitive impairment (PSCI) remains under-recognized in Sub-Saharan Africa (SSA), in part due to the lack of validated cognitive screening tools adapted to low-literacy populations. We aimed to validate the Identification of Dementia in Elderly Africans (IDEA) cognitive screen in SSA and assess its utility for detecting PSCI in Guinea and Cameroon. Methods: Normative IDEA scores were derived from a control cohort of healthy older adults in Conakry (Guinea) and Bafoussam (Cameroon). The tool was then applied to consecutive stroke patients from the same hospitals within one month of stroke onset. Demographic, clinical, and vascular risk profiles were collected. Between-group comparisons were performed using Welch’s *t*-tests and chi-square tests. Results: Among 91 healthy controls (median age: 64), the mean IDEA score was 12 ± 2.4. A cut-off of ≤7 (2 standard deviations below the mean) was defined for cognitive impairment. Among 111 stroke patients (median age: 65; mean NIHSS: 9.9 ± 5.8), the mean IDEA score was 9.6 ± 3.2, and 31 patients (28%) had scores ≤ 7. Stroke patients had significantly higher rates of hypertension and diabetes compared to controls. Conclusions: The IDEA screen appears to be a feasible and effective tool for detecting PSCI in SSA clinical settings. The 28% prevalence of cognitive impairment aligns with data from high-income countries, supporting the broader use of the IDEA to strengthen cognitive care pathways in SSA stroke populations.

## 1. Introduction

Stroke is a leading cause of disability, major neurocognitive disorder, and death worldwide. Post-stroke cognitive impairment (PSCI) affects up to 50% of stroke survivors, according to studies conducted in high-income countries [1,2]. Notably, PSCI is partly independent of the modified Rankin Scale used to assess functional outcomes [3], yet it significantly impacts patients’ daily lives [4]. Furthermore, PSCI limits independence, social participation [5], and is associated with increased rates of mortality, institutionalization, and depression [6].

Guidelines recommend routine cognitive screening for all stroke survivors [7], but this is rarely implemented, especially in low-resource settings. Limited awareness of PSCI’s long-term impact [5], combined with a lack of culturally and linguistically appropriate screening tools, contributes to low screening rates [7]. Emerging data suggest that PSCI may pose an even greater public health burden in low- and middle-income countries, particularly in Sub-Saharan Africa (SSA). Stroke incidence in SSA is estimated to be two- to threefold higher than in Western Europe and North America [8] and often affects individuals in their most productive decades of life. Cognitive sequelae following stroke therefore have substantial implications not only for individual patients but also for families and national productivity. Despite this, the epidemiology of PSCI in SSA remains poorly documented. Recent large-scale SSA studies support the urgency of addressing PSCI. A 2024 meta-analysis by Aytenew et al. pooled data from 1566 stroke survivors and reported a PSCI prevalence of 59.6%, with significant associations with age, education level, functional status, and stroke laterality [9]. Similarly, the CogFAST Nigeria study by Akinyemi et al. [10] found that nearly half of stroke survivors developed vascular cognitive impairment (VCI) within three months of stroke, with strong links to vascular risk factors, medial temporal atrophy, and lower education. Some other studies in SSA have reported prevalence rates comparable to those in high-income countries using the cognitive part of the Community Screening Instrument for Dementia (CSID), the mini-mental state examination (MMSE), and the Vascular Neuropsychological Battery (V-NB) [10], or even higher rates using the Montreal Cognitive Assessment (MoCA) [11,12]. However, screening tests like MocA yield high false-positive rates in culturally/linguistically diverse settings [13], underlining the need for validated cognitive screening tools adapted to low-literacy contexts hampers reliable measurement, screening, and follow-up. Interestingly, the Identification of Dementia in Elderly Africans (IDEA) cognitive screen was developed in rural Tanzania, specifically tailored for SSA populations, and has shown validity in detecting major neurocognitive disorders in both community and hospital settings in Tanzania and Nigeria [14,15].

Building on this context, we aimed to assess the performance of the IDEA screen in detecting PSCI among stroke survivors in Guinea and Cameroon and to evaluate its applicability in two Western African populations. Our goal was to determine whether the IDEA is a practical tool for routine cognitive screening in SSA stroke care.

## 2. Subjects and Methods

### 2.1. Subjects

Participants were recruited from two urban neurology departments: Ignace Deen Hospital in Conakry, Guinea, and Bafoussam Regional Hospital in Bafoussam, Cameroon. The stroke group included adult patients (≥40 years) with a clinical diagnosis of stroke according to WHO criteria, confirmed by CT imaging. Inclusion criteria for stroke patients were as follows: (1) first-ever or recurrent ischemic or hemorrhagic stroke; (2) ability to undergo cognitive screening within 30 days post-stroke; and (3) consent by the participant or caregiver. Exclusion criteria were as follows: history of pre-stroke dementia, severe aphasia or delirium precluding testing, or major psychiatric illness. The control group consisted of healthy older adults recruited from hospital visitors, community health workers, and relatives of patients. These individuals were age-matched as closely as feasible and screened to exclude any history of stroke, neurological or psychiatric conditions, or current cognitive complaints. Equal proportions were recruited from Guinea and Cameroon. This study was approved by institutional Ethics Committees. Each participant was given verbal information about this study and allowed to ask questions. Written informed consent was obtained by signature. An assent was sought from a close relative if the participant was unable to give consent, due to cognitive impairment.

### 2.2. Methods

Demographic variables (age, sex, literacy), vascular risk factors (hypertension, diabetes), and clinical information (stroke subtype, NIHSS score at admission) were collected from medical records and structured interviews. High blood pressure was defined as systolic blood pressure above 140 mmHg or reported hypertension on admission. Diabetes was defined as fasting glucose on admission ≥126 mg/dL [16]. Stroke subtypes were categorized as ischemic or hemorrhagic.

The IDEA cognitive screen includes six domains: abstraction, orientation, long-term memory, categorical verbal fluency, delayed recall, and visuospatial construction. It is scored from 0 to 15, with higher scores indicating better cognition. Items were administered in the local language by trained clinicians. A cut-off score for abnormal performance was derived from the control population: cognitive impairment was defined as a score ≤ 7, corresponding to more than 2 standard deviations below the control mean. This threshold reflects approximately the 2.5th percentile, a convention used in previous screening test studies on the MMSE [17], the MoCA [18], and the studies that validated IDEA norms [14,15]. IDEA testing in stroke patients was conducted within one month post-stroke. All assessments were completed in a quiet clinical setting and supervised by neurologists.

### 2.3. Statistical Analysis

All analyses were conducted using SPSS v26. Continuous variables were reported as means ± standard deviations or medians with interquartile ranges. Categorical variables were reported as frequencies and percentages. Welch’s *t*-test was chosen to account for unequal variances and sample sizes between groups [19]. Chi-square tests were used for categorical comparisons. A significance threshold of *p* < 0.05 was applied. Subgroup analysis (e.g., by age, sex, or comorbidities) was not performed due to sample size constraints.

## 3. Results

A total of 91 neurologically healthy individuals were included as controls. The mean age was 61 ± 12 years, with 52% male participants. Literacy rate was 79%, while hypertension and diabetes were present in 22% and 9.7% of subjects, respectively. The mean IDEA score in this group was 12 ± 2.4. Based on this distribution, a cut-off score of ≤7 (two standard deviations below the mean) was established to define probable cognitive impairment.

In the stroke group, 111 patients were included. Their mean age was 65 ± 9 years, and 50% were male. The literacy rate was 70%, and hypertension and diabetes were present in 70% and 49% of patients, respectively. Ischemic strokes accounted for 78% of cases. The mean NIHSS score at admission was 9.9 ± 5.8.

IDEA score distributions between groups (Figure 1B) were statistically compared using Welch’s t-test (t = 6.32, *p* < 0.001), and mean IDEA scores were significantly lower in stroke patients (9.6 ± 3.2) than in healthy controls (12 ± 2.4; *p* < 0.001). Additionally, 31 stroke patients (28%) and 5 healthy controls (5.5%) scored ≤7, indicating probable cognitive impairment (χ^2^ = 20.5, *p* < 0.001).

Group comparisons revealed no significant difference in sex distribution (χ^2^ = 0.08, *p* = 0.78) or literacy rates (χ^2^ = 2.45, *p* = 0.12) between groups. However, hypertension (χ^2^ = 46.6, *p* < 0.001) and diabetes (χ^2^ = 32.1, *p* < 0.001) were significantly more prevalent in the stroke group.

Participant characteristics and statistical comparisons are presented in Table 1.

Figure 1 illustrates the IDEA score distribution.

## 4. Discussion

Main findings from our study are that (i) the IDEA is adapted and effective to screen for cognitive disorders in Guinea and Cameroon and that (ii) using the IDEA cut-off, PSCI may affect up to 30% of post-stroke patients.

Those results from Guinean and Cameroonian cohorts are likely to be generalizable in similar settings in SSA. Our cohort of stroke patients that was included in this project matches the characteristics from prior large stroke cohorts from Guinea [20], Cameroon [21], South Africa [22], or the Stroke Investigative Research & Educational Network (SIREN) collaboration [23] in terms of age, sex, ischemic stroke proportion, and NIHSS severity. The prevalence of diabetes in our control population is similar to that reported in SSA [24]. The proportion of patients with elevated glucose levels in our stroke cohort is also close to the ~40% reported in acute stroke studies [25], part of which likely reflects stress hyperglycemia, rather than pre-existing diabetes, especially considering diabetes prevalence in other SSA stroke cohorts [24]. So, while limited by the sample sizes of our cohorts, the results from this report are likely to be generalizable to SSA healthy individuals as well as to SSA stroke patients in general. Similarly, our healthy cohort scored strikingly close performances compared to healthy individuals from Nigeria and Tanzania, in whom the IDEA was validated [14,15,26]. Our control cohort, comprising literate and low-literate older adults from Guinea and Cameroon, achieved a mean IDEA score of 12 ± 2.4, highly comparable to normative values reported in Tanzania and Nigeria by Gray et al. (2021) [26], who found median scores ranging from 10 to 13 depending on age, education, and sex. This similarity suggests that our control population is representative of wider SSA contexts and supports the use of a ≤7 cut-off for identifying cognitive impairment, as validated in previous IDEA studies. The IDEA tool thus shows promise as a generalizable screening instrument across SSA.

The 28% prevalence of PSCI detected using the IDEA is strikingly close to what has been reported in stroke survivors assessed with the Montreal Cognitive Assessment (MoCA) or Mini-Mental State Examination (MMSE) in high-income settings [2,27]. In contrast, recent SSA studies applying the MoCA reported much higher PSCI rates—up to 80 [11,12]. These discrepancies likely stem in part from the MoCA’s high false-positive rates when used in linguistically or culturally distinct populations without proper normative adjustments [13]. Supporting this, Masika et al. (2021) [28] demonstrated pronounced education-dependent floor effects when the MoCA was applied to rural Tanzanian participants. By contrast, the IDEA screen showed lower cultural bias and comparable diagnostic accuracy, even when directly compared to MoCA in the same study [28]. These findings reinforce the IDEA screen’s relevance and appropriateness in low-literacy SSA populations and support the generalizability of our results across similar regional settings.

Our results on PSCI prevalence are on the lower edge of those reported in a recent Sub-Saharan African (SSA) meta-analyses on post-stroke cognitive impairment (PSCI). Aytenew et al. (2024) reported a pooled PSCI prevalence of 59.6% across SSA [9]. This relatively lower rate of PSCI in Guinea may be attributed to the fact that many severe stroke cases likely failed to reach the hospital, due to limited accessibility. Barriers such as distance, transportation costs, and the lack of organized medical transport hinder timely admission. A 2014 survey conducted at the Neurology Ward of Ignace Deen Hospital revealed that only 2% of stroke patients arrived by ambulance, 46% used public transportation, 27% arrived by personal car, and the remainder relied on alternative or informal means [29]. While vascular risk factors like hypertension and diabetes can modestly affect cognitive performance in their own right, higher rates of diabetes and HBP in our stroke patients could have negatively biased the results of IDEA screening independently of the stroke event. Diabetic and HBP patients tend to perform poorer than matched controls on neuropsychological testing [30,31,32]. However, those lower performances are subtle and inconsistent across studies and do not significantly impact cognitive screening [30,31,32], suggesting that the bulk of PSCI detected by the IDEA are related to the stroke event. Still, although our sample included diabetic individuals in the control group, further stratification based on comorbidities is warranted in future studies to refine comparisons and better elucidate the potential influence of hyperglycemia on PSCI.

Importantly, in addition to general prevalence estimates, emerging evidence highlights that several clinical and sociodemographic factors can help predict the risk of PSCI and guide targeted intervention strategies. For instance, a study by Ojagbemi et al. (2021) in the Nigerian cohort of the SIREN project showed that pre-stroke cognitive decline, as measured by informant-based tools, was independently associated with an increased risk of developing post-stroke dementia [33]. This underscores the need for the early identification of at-risk individuals through caregiver-based screening tools or cognitive history at the point of care. Similarly, a recent prospective study from Tanzania by Alphonce et al. (2024) identified age > 60 years, low education, left hemispheric stroke, and low functional independence at discharge as strong predictors of PSCI at follow-up [11]. Incorporating such predictors into post-stroke care pathways could facilitate early cognitive screening with context-adapted tools like the IDEA, more personalized rehabilitation planning, and the prioritization of neurocognitive support for the most vulnerable stroke survivors.

Despite being field-friendly, screening tools cannot replace full-length neuropsychological test batteries that have higher sensitivities to detect and characterize cognitive impairments, as shown by the rates of PSCI amounting to 50% when more comprehensive evaluations are used [1,34]. Still, dedicated bedside assessment tests for PSCI, such as the Oxford Cognitive Screen (OCS), have gained acceptance for the assessment of cognitive impairments with robust validity and reliability [35]. The OCS is a neurobehavioral battery, like the IDEA, that allows a rapid assessment of several cognitive domains (i.e., language, memory, attention, calculation, and praxis) and shows high predictive abilities for long-term stroke functional outcome [36]. This shows that adapted bedside cognitive tests can help predict stroke functional outcomes and orient patient care and rehabilitation. In the SSA context where full neuropsychological test batteries and neuropsychologists are scarce and/or inaccessible, the IDEA could stand as a pragmatic screening test to identify individuals who may be cognitively impacted by a stroke in SSA and help design both public health and rehabilitation strategies as well as attract attention to a relevant and under-recognized consequence of stroke in Africa.

Our study has several limitations: First, the absence of an a priori sample size calculation, which was not feasible given the exploratory nature and logistical constraints of recruitment in SSA clinical settings. However, a post hoc power analysis based on the observed difference in IDEA scores between stroke patients (M = 9.6, SD = 3.2) and controls (M = 12.0, SD = 2.4) yielded a Cohen’s *d* of 0.83, indicating a large effect size [37]. With this effect size and our sample size (n = 202; 111 stroke, 91 control), this study achieved over 95% power at α = 0.05 using a two-tailed Welch’s *t*-test. This suggests that the sample was sufficiently powered to detect meaningful group differences, despite the absence of formal prospective power planning. Second, feasibility, effectiveness, and the cultural relevance of the IDEA screen were not assessed as formal study outcomes. Although the tool was well tolerated, easily administered, and showed expected discrimination between groups, these aspects were not explicitly predefined or systematically evaluated. Future studies should include structured feasibility metrics, qualitative feedback from patients and clinicians, and cross-cultural validation frameworks to formally assess these important implementation dimensions. Finally, this study lacks the evaluation of concurrent cognitive assessments, which precluded the direct comparison of IDEA performance with other validated screening instruments such as the MoCA or MMSE. Future studies should incorporate the parallel administration of multiple tools to allow for convergent validity analyses and enhance interpretability.

## 5. Conclusions

Our findings underscore the relevance and feasibility of implementing routine cognitive screening for stroke patients in SSA using the IDEA tool. Doing so would not only support early intervention and rehabilitation but also bring attention to a major yet underrecognized consequence of stroke in Africa.

## Figures and Tables

**Figure 1 brainsci-15-00543-f001:**
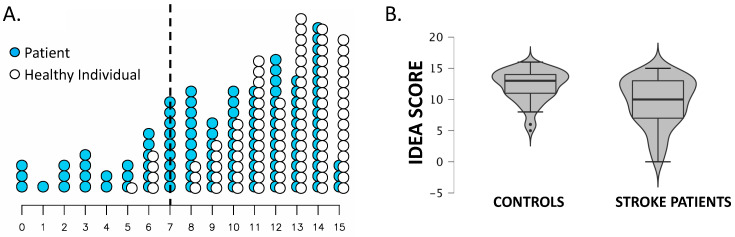
Panel (**A**) illustrates the distribution of IDEA scores. The black dotted line cuts off values under 7 that are considered to be abnormally low. Panel (**B**): Violin plot of the IDEA score distribution in healthy subjects and post-stroke individuals.

**Table 1 brainsci-15-00543-t001:** Participant characteristics and statistical comparisons.

Variable	Stroke Patients (*n* = 111)	Healthy Individuals (*n* = 91)	*p*-Value
Age (years), median [range]	65 [41–88]	64 [22–80]	<0.01
Male, %	50%	52%	0.487
Literacy, %	70%	79%	0.118
Hypertension (HBP), %	70%	22%	<0.01
Diabetes, %	49%	9.7%	<0.01
Ischemic stroke, %	78%	—	—
NIHSS, mean ± SD	9.9 ± 5.8	—	—
IDEA score, mean ± SD	9.6 ± 3.2	12 ± 2.4	<0.01
IDEA ≤ 7, n (%)	31 (28%)	5 (5.5%)	<0.01

Table legend: HBP: high blood pressure; NIHSS: National Institutes of Health Stroke Scale; SD: standard deviation; IDEA: Identification of Dementia in Elderly Africans cognitive screen.

## Data Availability

The original contributions presented in this study are included in this article/Appendix A. Further inquiries can be directed to the corresponding author(s).

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
