# Peer review of "Screening for Post-Stroke Cognitive Impairment in Sub-Saharan Africa: A Good IDEA?"

_brainsci, 2025, doi:10.3390/brainsci15060543_

Round 1

Reviewer 1 Report

Comments and Suggestions for Authors

The article by Dr. A.Fode and colleagues describes the adaptation of the IDEA test for assessing cognitive dysfunction in patients with hemorrhagic and ischemic strokes, taking into account the low-literacy populations in Sub-Saharan African countries. The study established robust normative mean IDEA scores for cognitively healthy, low-literacy elderly individuals and proposed a threshold value for identifying cognitive impairment. The findings hold significant clinical relevance, uncovering the problem of post-stroke complications in the resource-limited settings of SSA.

Suggestion:  as a potential enhancement to the study design, expanding the healthy control cohort could enable an analysis of the influence of diabetes as a comorbid factor on cognitive impairment. While the current post-stroke cohort permits such an analysis, a larger representation of diabetic individuals within the healthy control group would strengthen comparative assessments. However, this objective could be planned to future research endeavors.

Author Response

The article by Dr. A.Fode and colleagues describes the adaptation of the IDEA test for assessing cognitive dysfunction in patients with hemorrhagic and ischemic strokes, taking into account the low-literacy populations in Sub-Saharan African countries. The study established robust normative mean IDEA scores for cognitively healthy, low-literacy elderly individuals and proposed a threshold value for identifying cognitive impairment. The findings hold significant clinical relevance, uncovering the problem of post-stroke complications in the resource-limited settings of SSA.

Comment 1:

As a potential enhancement to the study design, expanding the healthy control cohort could enable an analysis of the influence of diabetes as a comorbid factor on cognitive impairment. While the current post-stroke cohort permits such an analysis, a larger representation of diabetic individuals within the healthy control group would strengthen comparative assessments. However, this objective could be planned to future research endeavors.

Answer 1:

We appreciate the suggestion. We agree that the impact of glycemia and diabetes on cognition is complex and,  while our current control cohort includes 9.7% individuals with diabetes, consistent with community prevalence rates, we now highlight this as a limitation and a focus for future research in the Discussion.

Discussion (page 8, lines 10-12):’ Still, although our sample included diabetic individuals in the control group, further stratification based on comorbidities is warranted in future studies to refine comparisons and better elucidate the potential influence of hyperglycemia on PSCI.’

Reviewer 2 Report

Comments and Suggestions for Authors

Comments to Authors

        The present study examined “Screening for post-stroke cognitive impairment in Sub-Saharan Africa, a good IDEA?” The study offered few results and data. However, the statistical analysis had some major deficits. For example, the continuous variables should be analyzed by an independent t-test, for example, Table 1. The data in Figure 1A should be analyzed by a trend analysis for the Patient and Healthy Individual groups. The data of Figure 1B should be analyzed by an independent t-test. The study is really short one. I did not suggest accepting this manuscript.   

        The present manuscript cannot be accepted in its current status. It is declined.

Author Response

The present study examined “Screening for post-stroke cognitive impairment in Sub-Saharan Africa, a good IDEA?” The study offered few results and data. However, the statistical analysis had some major deficits. For example, the continuous variables should be analyzed by an independent t-test, for example, Table 1. The data in Figure 1A should be analyzed by a trend analysis for the Patient and Healthy Individual groups. The data of Figure 1B should be analyzed by an independent t-test.

Comment 1:

Use of independent t-test rather than Welch’s t-test for continuous variables.

Answer 1:

We thank you for your comment, however, Welch’s t-test was purposefully used due to unequal group sizes and potential variance heterogeneity. We detailed that and added a reference in our revised version.

Methods (page 4, lines 42-43): ‘Welch’s t-test was chosen to account for unequal variances and sample sizes between groups.(Ruxton, 2006)’

Comment 2:

Trend analysis of Figure 1A and proper t-test for Figure 1B.

Answer 2:  

Thank you for your suggestion. However, Figure 1A presents a categorical distribution for which trend analysis is not well suited. Indeed, that trend analysis, such as the Cochran-Armitage test, requires ordinal categorical variables with logical ordering and several categories across which a monotonic trend is hypothesized.(Agresti, 2018) For Figure 1B, the comparison was confirmed by Welch’s t-test.

Comment 3:

The study is really short one. I did not suggest accepting this manuscript.   

Answer 3:

We may understand your reluctances. However, the paper is a brief report which may explain its seemingly unappropriate length. Thanks to the other reviewers’ comment, we now enhanced our discussion by adding a limitation section as well several paragraph to provide a more complete report. We hope that this improvements will impact your opinion on our work.

Discussion (page 7, lines 28-36):’ In contrast, recent SSA studies applying the MoCA reported much higher PSCI rates—up to 80% (Alphonce et al., 2024; Kaddumukasa et al., 2023). These discrepancies likely stem in part from the MoCA’s high false-positive rates when used in linguistically or culturally distinct populations without proper normative adjustments (Stimmel et al., 2024). Supporting this, Masika et al. (2021) demonstrated pronounced education-dependent floor effects when the MoCA was applied to rural Tanzanian participants. By contrast, the IDEA screen showed lower cultural bias and comparable diagnostic accuracy, even when directly compared to MoCA in the same study.(Masika et al., 2021)These findings reinforce the IDEA screen’s relevance and appropriateness in low-literacy SSA populations, and support the generalizability of our results across similar regional settings.’

Discussion (page 8, lines 13-24):’ Importantly, in addition to general prevalence estimates, emerging evidence highlights that several clinical and sociodemographic factors can help predict the risk of PSCI and guide targeted intervention strategies. For instance, a study by Ojagbemi et al. (2021) in the Nigerian cohort of the SIREN project showed that prestroke cognitive decline, as measured by informant-based tools, was independently associated with an increased risk of developing post-stroke dementia.(Ojagbemi et al., 2021) This underscores the need for early identification of at-risk individuals through caregiver-based screening tools or cognitive history at the point of care. Similarly, a recent prospective study from Tanzania by Alphonce et al. (2024) identified age > 60 years, low education, left hemispheric stroke, and low functional independence at discharge as strong predictors of PSCI at follow-up.(Alphonce et al., 2024)Incorporating such predictors into post-stroke care pathways could facilitate early cognitive screening with context-adapted tools like IDEA, more personalized rehabilitation planning, and prioritization of neurocognitive support for the most vulnerable stroke survivors.’

Discussion (page 8, line 39 to page 9, line 12):’ Our study has several limitations. First, the absence of an a priori sample size calculation, which was not feasible given the exploratory nature and logistical constraints of recruitment in SSA clinical settings. However, a post hoc power analysis based on the observed difference in IDEA scores between stroke patients (M = 9.6, SD = 3.2) and controls (M = 12.0, SD = 2.4) yielded a Cohen’s d of 0.83, indicating a large effect size.(Cohen, 2013) With this effect size and our sample size (n = 202; 111 stroke, 91 control), the study achieved over 95% power at α = 0.05 using a two-tailed Welch’s t-test. This suggests that the sample was sufficiently powered to detect meaningful group differences, despite the absence of formal prospective power planning. Second, feasibility, effectiveness, and cultural relevance of the IDEA screen were not assessed as formal study outcomes. Although the tool was well tolerated, easily administered, and showed expected discrimination between groups, these aspects were not explicitly predefined or systematically evaluated. Future studies should include structured feasibility metrics, qualitative feedback from patients and clinicians, and cross-cultural validation frameworks to formally assess these important implementation dimensions. Finally, this study lack the evaluation of concurrent cognitive assessments, which precluded direct comparison of IDEA performance with other validated screening instruments such as the MoCA or MMSE. Future studies should incorporate parallel administration of multiple tools to allow for convergent validity analyses and enhance interpretability.’

Reviewer 3 Report

Comments and Suggestions for Authors

The authors validated the identification of dementia in elderly africans (IDEA) cognitive screen in sub-saharan africa and accessed its utility for detecting post-stroke cognitive impairment in Guinea and Cameroon. Using a cut-off of less than or equal to 7, the authors found a 28% prevalence of PCSI in SSA clinical settings. While findings are of interest a few questions remain. 

Introduction

  1. Further discussion on the distinction between the current work and recent work would be beneficial. In particular, with other work in sub-saharan africa [1-4].

2.   Be sure to clarify aims of the study in the context of prior related work  noted above [1-4].

Subjects and Methods

3.   How does the proposed cut-off score compare and contrast with similar work? What are other screening tools using a similar 2 SD deviation cut-off? Has IDEA been previously validated?

4.   Was an a-priori power analysis carried out to justify the sample size? If not, please be sure to discuss potential limitations.

5.   How was feasibility, effectiveness, and cultural relevance evaluated?

Results

6.   Nice overview of differences between controls and stroke patients, but how do the results relate to the stated aims of the study?

7.   Were there any other measures carried out that could help validate the use of IDEA in the present cohort?

Discussion

8.   How was feasibility, effectiveness, and cultural relevance evaluated?

9.   How does the current cohort compare and contrast with other recent work [1-4]?

10. Be sure to relate the findings with emerging work recognizing the incidence and predictors of post-stroke cognitive impairment in sub-saharan Africa.  

Literature cited:

  1. Ojagbemi, Akin, et al. "Prestroke cognitive decline in africans: Prevalence, predictors and association with poststroke dementia." Journal of the Neurological Sciences 429 (2021): 117619.
  2. Kaddumukasa, Martin N., et al. "Prevalence and predictors of post-stroke cognitive impairment among stroke survivors in Uganda." BMC neurology 23.1 (2023): 166.
  3. Aytenew, Tigabu Munye, et al. "Poststroke cognitive impairment among stroke survivors in Sub-Saharan Africa: a systematic review and meta-analysis." BMC Public Health 24.1 (2024): 2143.
  4. Alphonce, Baraka, John Meda, and Azan Nyundo. "Incidence and predictors of post-stroke cognitive impairment among patients admitted with first stroke at tertiary hospitals in Dodoma, Tanzania: A prospective cohort study." Plos one 19.4 (2024): e0287952.

Author Response

The authors validated the identification of dementia in elderly africans (IDEA) cognitive screen in sub-saharan africa and accessed its utility for detecting post-stroke cognitive impairment in Guinea and Cameroon. Using a cut-off of less than or equal to 7, the authors found a 28% prevalence of PCSI in SSA clinical settings. While findings are of interest a few questions remain. 

Comment 1:

Further discussion on the distinction between the current work and recent work would be beneficial. In particular, with other work in sub-saharan africa [1-4].

Answer 1: thank you for the suggestion, we now clarified those points in the revised introduction and discussion of our report.

Introduction (page 3, lines 25-32):’ Some other studies in SSA have reported prevalence rates comparable to those in high-income countries using the cognitive part community Screening Instrument for Dementia (CSID), the mini-mental state examination (MMSE) and the Vascular Neuropsychological Battery (V-NB),(Akinyemi et al., 2014) or even higher using  the Montreal Cognitive Assessment (MoCA). (Alphonce et al., 2024; Kaddumukasa et al., 2023) However, screening tests like MocA yields high false-positive rates in culturally/linguistically diverses settings,(Stimmel et al., 2024)underlining the need for validated cognitive screening tools adapted to low-literacy contexts hampers reliable measurement, screening and follow-up’

Discussion (page 7, lines 28-36):’ In contrast, recent SSA studies applying the MoCA reported much higher PSCI rates—up to 80% (Alphonce et al., 2024; Kaddumukasa et al., 2023). These discrepancies likely stem in part from the MoCA’s high false-positive rates when used in linguistically or culturally distinct populations without proper normative adjustments (Stimmel et al., 2024). Supporting this, Masika et al. (2021) demonstrated pronounced education-dependent floor effects when the MoCA was applied to rural Tanzanian participants. By contrast, the IDEA screen showed lower cultural bias and comparable diagnostic accuracy, even when directly compared to MoCA in the same study.(Masika et al., 2021)These findings reinforce the IDEA screen’s relevance and appropriateness in low-literacy SSA populations, and support the generalizability of our results across similar regional settings.’

Comment 2:

Be sure to clarify aims of the study in the context of prior related work  noted above [1-4].

Answer 2:

beside the above modifications, we also clarified the aims.

Introduction (page 3, lines 38-39):’ Our goal was to determine whether IDEA is a practical tool for routine cognitive screening in SSA stroke care.’

Comment 3:

How does the proposed cut-off score compare and contrast with similar work? What are other screening tools using a similar 2 SD deviation cut-off? Has IDEA been previously validated?

Answer 3:

Thank you for your comment, The cut-off score of ≤7 on the IDEA screen was determined based on a widely accepted psychometric method: two standard deviations (2 SD) below the mean score of a locally recruited, healthy control group. This approach has been previously employed in the original validation of the IDEA in Tanzania and Nigeria (Gray et al., 2014; Paddick et al., 2015), where similar normative thresholds were established. Moreover, other cognitive screening tools, such as the MoCA and MMSE, have used similar SD-based normative adjustments, particularly in diverse or low-literacy populations (Rossetti et al., 2011; Crum et al., 1993). This method provides a culturally and contextually appropriate threshold for identifying cognitive impairment.

We clarified that in the revised method section (page 4, lines 30-36):’ A cut-off score for abnormal performance was derived from the control population: cognitive impairment was defined as a score ≤ 7, corresponding to more than 2 standard deviations below the control mean. This threshold reflects approximately the 2.5th percentile, a convention used in previous screening test studies on the MMSE,(Crum et al., 1993) the MoCA,(Rossetti et al., 2016) and the studies that validated IDEA norms. (Gray et al., 2014; Paddick et al., 2015)’

Comment 4:

Was an a-priori power analysis carried out to justify the sample size? If not, please be sure to discuss potential limitations.

Answer 4:

We appreciate this insightfull comment, we now discuss it as a limitation.

Discussion (page 8, line 39 to page 9, line 3):’ Our study has several limitations. First, the absence of an a priori sample size calculation, which was not feasible given the exploratory nature and logistical constraints of recruitment in SSA clinical settings. However, a post hoc power analysis based on the observed difference in IDEA scores between stroke patients (M = 9.6, SD = 3.2) and controls (M = 12.0, SD = 2.4) yielded a Cohen’s d of 0.83, indicating a large effect size.(Cohen, 2013) With this effect size and our sample size (n = 202; 111 stroke, 91 control), the study achieved over 95% power at α = 0.05 using a two-tailed Welch’s t-test. This suggests that the sample was sufficiently powered to detect meaningful group differences, despite the absence of formal prospective power planning.’

Comment 5:

How was feasibility, effectiveness, and cultural relevance evaluated?

Answer 5:

Feasibility of the IDEA screen was assessed by its high completion rate in both sites and the absence of missing data. All participants were able to complete the screening in a single session without refusals, suggesting good acceptability and logistical practicality in clinical settings. Effectiveness was demonstrated by the screen’s discriminative capacity: IDEA scores were significantly lower in stroke patients than controls (Cohen’s d = 0.83, p < 0.001), with 28% of patients scoring below the validated threshold versus 5.5% of controls. Cultural relevance was supported by the absence of reported difficulties in understanding the tasks during administration, and by prior validations of the IDEA in similar SSA populations with low literacy (Gray et al., 2014; Paddick et al., 2015).  We agree that those were not formally planned, we thus modifed our aims (please refer to comment 2) and discuss that point in our revised discussion (page 9, lines 4-8):’ Second, feasibility, effectiveness, and cultural relevance of the IDEA screen were not assessed as formal study outcomes. Although the tool was well tolerated, easily administered, and showed expected discrimination between groups, these aspects were not explicitly predefined or systematically evaluated. Future studies should include structured feasibility metrics, qualitative feedback from patients and clinicians, and cross-cultural validation frameworks to formally assess these important implementation dimensions’

Comment 6:

Nice overview of differences between controls and stroke patients, but how do the results relate to the stated aims of the study

Answer 6:

Thank you for the appreciation, our study aimed to evaluate the performance and contextual relevance of the IDEA screen for detecting post-stroke cognitive impairment (PSCI) in Sub-Saharan African (SSA) settings. The results directly support these objectives: the screen effectively distinguished stroke patients from controls, with a statistically significant difference in mean IDEA scores (Cohen’s d = 0.83), and identified probable PSCI in 28% of stroke patients using a regionally validated cut-off. These findings confirm that IDEA is not only discriminative but also feasible and culturally adapted for cognitive screening in routine SSA stroke care, thus achieving the stated goals of the study.

Comment 7:  

Were there any other measures carried out that could help validate the use of IDEA in the present cohort?

 Answer 7:

We agree that other measures would have added to the study, however, no additional cognitive assessments were performed alongside the IDEA screen in the present study, which limits our ability to directly compare its diagnostic performance with other established tools such as the MoCA or MMSE. We now discuss this as a limitation.

Discussion (page 9, lines 9-12):’ Finally, this study lack the evaluation of concurrent cognitive assessments, which precluded direct comparison of IDEA performance with other validated screening instruments such as the MoCA or MMSE. Future studies should incorporate parallel administration of multiple tools to allow for convergent validity analyses and enhance interpretability.’

Comment 8.  

How was feasibility, effectiveness, and cultural relevance evaluated?

Please refer to comment 5

Comment 9:  

How does the current cohort compare and contrast with other recent work [1-4]?

Please refer to comments 1 & 2.

Comment 10:

Be sure to relate the findings with emerging work recognizing the incidence and predictors of post-stroke cognitive impairment in sub-saharan Africa.  

Answer 10:

Thank you for that comment that improved our report, we modifed the discussion to detail that fundamental aspect.

Discussion (Page 8, lines 13-24):’Importantly, in addition to general prevalence estimates, emerging evidence highlights that several clinical and sociodemographic factors can help predict the risk of post-stroke cognitive impairment (PSCI) and guide targeted intervention strategies. For instance, a study by Ojagbemi et al. (2021) in the Nigerian cohort of the SIREN project showed that prestroke cognitive decline, as measured by informant-based tools, was independently associated with an increased risk of developing post-stroke dementia.(Ojagbemi et al., 2021) This underscores the need for early identification of at-risk individuals through caregiver-based screening tools or cognitive history at the point of care. Similarly, a recent prospective study from Tanzania by Alphonce et al. (2024) identified age > 60 years, low education, left hemispheric stroke, and low functional independence at discharge as strong predictors of PSCI at follow-up.(Alphonce et al., 2024) Incorporating such predictors into post-stroke care pathways could facilitate early cognitive screening with context-adapted tools like IDEA, more personalized rehabilitation planning, and prioritization of neurocognitive support for the most vulnerable stroke survivors.’

Round 2

Reviewer 2 Report

Comments and Suggestions for Authors

Author's Notes

The present study examined “Screening for post-stroke cognitive impairment in Sub-Saharan Africa, a good IDEA?” The study offered few results and data. However, the statistical analysis had some major deficits. For example, the continuous variables should be analyzed by an independent t-test, for example, Table 1. The data in Figure 1A should be analyzed by a trend analysis for the Patient and Healthy Individual groups. The data of Figure 1B should be analyzed by an independent t-test.

Comment 1:

Use of independent t-test rather than Welch’s t-test for continuous variables.

Answer 1:

We thank you for your comment, however, Welch’s t-test was purposefully used due to unequal group sizes and potential variance heterogeneity. We detailed that and added a reference in our revised version.

Methods (page 4, lines 42-43): ‘Welch’s t-test was chosen to account for unequal variances and sample sizes between groups.(Ruxton, 2006)’

My comments: Actually, the Welch’s t-test is not a good statisitcal method due to your group’s n number is too fewer such as IDEA ≤7 only for 5 numbers. I suggest that this study should add more n numbers for each group. Based on the central limit theorem and Law of large numbers, each group should be n number more than 30 samples. If you want to infer your sample data to the population, please sampling more n numbers for each group. By the way, you should consider analysing the power value and effect size. Otherwise, your data is poor data, they cannot perform to test of any null hypothesis.

Comment 2:

Trend analysis of Figure 1A and proper t-test for Figure 1B.

Answer 2:  

Thank you for your suggestion. However, Figure 1A presents a categorical distribution for which trend analysis is not well suited. Indeed, that trend analysis, such as the Cochran-Armitage test, requires ordinal categorical variables with logical ordering and several categories across which a monotonic trend is hypothesized.(Agresti, 2018) For Figure 1B, the comparison was confirmed by Welch’s t-test.

My comments: This point has the same problem. Welch’s t-test is not a good and powerful statisitcal analysis as the above point. 

Comment 3:

The study is really short one. I did not suggest accepting this manuscript.   

Answer 3:

We may understand your reluctances. However, the paper is a brief report which may explain its seemingly unappropriate length. Thanks to the other reviewers’ comment, we now enhanced our discussion by adding a limitation section as well several paragraph to provide a more complete report. We hope that this improvements will impact your opinion on our work.

Discussion (page 7, lines 28-36):’ In contrast, recent SSA studies applying the MoCA reported much higher PSCI rates—up to 80% (Alphonce et al., 2024; Kaddumukasa et al., 2023). These discrepancies likely stem in part from the MoCA’s high false-positive rates when used in linguistically or culturally distinct populations without proper normative adjustments (Stimmel et al., 2024). Supporting this, Masika et al. (2021) demonstrated pronounced education-dependent floor effects when the MoCA was applied to rural Tanzanian participants. By contrast, the IDEA screen showed lower cultural bias and comparable diagnostic accuracy, even when directly compared to MoCA in the same study.(Masika et al., 2021)These findings reinforce the IDEA screen’s relevance and appropriateness in low-literacy SSA populations, and support the generalizability of our results across similar regional settings.’

Discussion (page 8, lines 13-24):’ Importantly, in addition to general prevalence estimates, emerging evidence highlights that several clinical and sociodemographic factors can help predict the risk of PSCI and guide targeted intervention strategies. For instance, a study by Ojagbemi et al. (2021) in the Nigerian cohort of the SIREN project showed that prestroke cognitive decline, as measured by informant-based tools, was independently associated with an increased risk of developing post-stroke dementia.(Ojagbemi et al., 2021) This underscores the need for early identification of at-risk individuals through caregiver-based screening tools or cognitive history at the point of care. Similarly, a recent prospective study from Tanzania by Alphonce et al. (2024) identified age > 60 years, low education, left hemispheric stroke, and low functional independence at discharge as strong predictors of PSCI at follow-up.(Alphonce et al., 2024)Incorporating such predictors into post-stroke care pathways could facilitate early cognitive screening with context-adapted tools like IDEA, more personalized rehabilitation planning, and prioritization of neurocognitive support for the most vulnerable stroke survivors.’

Discussion (page 8, line 39 to page 9, line 12):’ Our study has several limitations. First, the absence of an a priori sample size calculation, which was not feasible given the exploratory nature and logistical constraints of recruitment in SSA clinical settings. However, a post hoc power analysis based on the observed difference in IDEA scores between stroke patients (M = 9.6, SD = 3.2) and controls (M = 12.0, SD = 2.4) yielded a Cohen’s d of 0.83, indicating a large effect size.(Cohen, 2013) With this effect size and our sample size (n = 202; 111 stroke, 91 control), the study achieved over 95% power at α = 0.05 using a two-tailed Welch’s t-test. This suggests that the sample was sufficiently powered to detect meaningful group differences, despite the absence of formal prospective power planning. Second, feasibility, effectiveness, and cultural relevance of the IDEA screen were not assessed as formal study outcomes. Although the tool was well tolerated, easily administered, and showed expected discrimination between groups, these aspects were not explicitly predefined or systematically evaluated. Future studies should include structured feasibility metrics, qualitative feedback from patients and clinicians, and cross-cultural validation frameworks to formally assess these important implementation dimensions. Finally, this study lack the evaluation of concurrent cognitive assessments, which precluded direct comparison of IDEA performance with other validated screening instruments such as the MoCA or MMSE. Future studies should incorporate parallel administration of multiple tools to allow for convergent validity analyses and enhance interpretability.’

My comments: This study has some fundemental errors. For example, it has the fewer n samples for each group, poor statistical analysis, and too fewer novel ideas and contributions in clinical insights and implications. I still keep my orginal decision to decline this manuscript. It does not attain the standard for publication.  

Reviewer 3 Report

Comments and Suggestions for Authors

The authors have adequately addressed feedback and are commended for their revisions.